# Pyridostigmine Treatment Significantly Alleviates Isoprenaline-Induced Chronic Heart Failure in Rats

**DOI:** 10.3390/ijms26146892

**Published:** 2025-07-17

**Authors:** Sonja T. Marinković, Tanja Sobot, Žana M. Maksimović, Ðorđe Ðukanović, Snežana Uletilović, Nebojša Mandić-Kovačević, Sanja Jovičić, Milka Matičić, Milica Gajić Bojić, Aneta Stojmenovski, Anđela Bojanić, Ranko Škrbić, Miloš P. Stojiljković

**Affiliations:** 1Centre for Biomedical Research, Faculty of Medicine, University of Banja Luka, 78000 Banja Luka, The Republic of Srpska, Bosnia and Herzegovina; sonja.trbojevic@med.unibl.org (S.T.M.); zana.maksimovic@med.unibl.org (Ž.M.M.); djordje.djukanovic@med.unibl.org (Ð.Ð.); snezana.uletilovic@med.unibl.org (S.U.); nebojsa.mandic-kovacevic@med.unibl.org (N.M.-K.); sanja.jovicic@med.unibl.org (S.J.); milka.maticic@med.unibl.org (M.M.); milica.gajic@med.unibl.org (M.G.B.); aneta.stojmenovski@med.unibl.org (A.S.); andjela.bojanic@med.unibl.org (A.B.); ranko.skrbic@med.unibl.org (R.Š.); 2Pediatric Clinic, University Clinical Centre of the Republic of Srpska, 78000 Banja Luka, The Republic of Srpska, Bosnia and Herzegovina; 3Department of Physiology, Faculty of Medicine, University of Banja Luka, 78000 Banja Luka, The Republic of Srpska, Bosnia and Herzegovina; tanja.sobot@med.unibl.org; 4Department of Pharmacology, Toxicology and Clinical Pharmacology, Faculty of Medicine, University of Banja Luka, 78000 Banja Luka, The Republic of Srpska, Bosnia and Herzegovina; 5Department of Pharmaceutical Chemistry, Faculty of Medicine, University of Banja Luka, 78000 Banja Luka, The Republic of Srpska, Bosnia and Herzegovina; 6Department of Medical Biochemistry and Chemistry, Faculty of Medicine, University of Banja Luka, 78000 Banja Luka, The Republic of Srpska, Bosnia and Herzegovina; 7Department of Pharmacy, Faculty of Medicine, University of Banja Luka, 78000 Banja Luka, The Republic of Srpska, Bosnia and Herzegovina; 8Department of Histology and Embryology, Faculty of Medicine, University of Banja Luka, 78000 Banja Luka, The Republic of Srpska, Bosnia and Herzegovina; 9Academy of Sciences and Arts of the Republic of Srpska, 78000 Banja Luka, The Republic of Srpska, Bosnia and Herzegovina; 10Department of Pathologic Physiology, I.M. Sechenov First Moscow State Medical University, 119435 Moscow, Russia

**Keywords:** pyridostigmine, acetylcholinesterase, chronic heart failure, rat, oxidative stress, cholinergic receptor

## Abstract

Autonomic imbalance is one of the major pathological disturbances in chronic heart failure (CHF). Additionally, enhanced oxidative stress and inflammation are considered to be the main contributors to the disease progression. A growing body of evidence suggests cholinergic stimulation as a potential therapeutic approach in CHF, since it corrects the autonomic imbalance and alters the inflammatory response via the cholinergic anti-inflammatory pathway. Although previous research has provided some insights into the potential mechanisms behind these effects, there is a gap in knowledge regarding different cholinergic stimulation methods and their specific mechanisms of action. In the present study, an isoprenaline model (5 mg/kg/day s.c. for 7 days, followed by 4 weeks of CHF development) was used. Afterwards, rats received pyridostigmine (22 mg/kg/day in tap water for 14 days) or no treatment. Pyridostigmine treatment prevented the progression of CHF, decreasing chamber wall thinning (↑ PWDd, ↑ PWDs) and left ventricle dilatation (↓ LVIDd, ↓ LVIDs), thus improving cardiac contractile function (↑ EF). Additionally, pyridostigmine improved antioxidative status (↓ TBARS, ↓ NO_2_^−^; ↑ CAT, ↑ GSH) and significantly reduced cardiac fibrosis development, confirmed by pathohistological findings and biochemical marker reduction (↓ MMP2, ↓ MMP9). However, further investigations are needed to fully understand the exact cellular mechanisms involved in the CHF attenuation via pyridostigmine.

## 1. Introduction

The autonomic nervous system (ANS), with its two branches—the sympathetic (SNS) and parasympathetic nervous systems (PNS)—is responsible for the unconscious control of the body’s homeostasis, both in physiologic and pathologic conditions. In the cardiovascular system, the ANS regulates vital functions such as heart rate, cardiac contractility, and vascular tone. In physiologic conditions, the two antagonistic branches maintain a delicate and balanced relationship. However, in the case of chronic heart failure (CHF), a multi-aetiological condition characterised by an impairment in the heart’s ability to fill with and pump blood, the autonomic balance is altered [1,2]. CHF is characterised by the overactivity of the SNS and parasympathetic withdrawal [2,3], because of which the longstanding recommendations for CHF treatment are based on the reduction in sympathetic effects. Thus, beta-blockers and renin–angiotensin–aldosterone system (RAAS) inhibitors are still the gold standard for CHF treatment [3]. However, this therapeutic regimen is not without faults. A limitation in the use of beta-blockers is their negative inotropic effect, which can be explained by the anatomy of the PNS of the heart. The SNS innervates both atria and ventricles, because of which beta-blockers may not be appropriate in decompensated patients [4,5]. On the other hand, parasympathetic modulation as an alternative or additional therapeutic strategy has not yet been fully investigated, although this topic has been gaining momentum in the last decade.

Electrical vagal nerve stimulation was the first to show a marked improvement in the long-term survival of rats with experimentally induced CHF. The effects were explained by cardiac contractile function preservation and cardiac remodelling reduction [6,7]. Although this treatment has also shown promising effects in clinical studies, its invasiveness and cost limitations have deterred further investigations. Non-invasive methods to increase the local concentration of acetylcholine (ACh), the vagal neurotransmitter, especially via the acetylcholinesterase (AChE) inhibition, have proven to be more suitable [8,9,10,11]. Two of the most widely used medications are reversible AChE inhibitors, donepezil and pyridostigmine. As both a central and peripheral AChE inhibitor, donepezil is used to increase central ACh levels as a treatment for some neurodegenerative diseases, such as Alzheimer’s disease and vascular dementia. On the other hand, pyridostigmine acts only as a peripheral AChE inhibitor and has found application in myasthenia gravis patients. AChE inhibitors have been shown to improve autonomic regulation, increase heart rate variability, and increase baroreflex sensitivity in acute myocardial infarction (AMI) rats [1,5,12,13]. Pyridostigmine administration in AMI rats led to the preservation of connexin 43, thus protecting against ischaemia-induced arrhythmias. Anti-inflammatory effects, oxidative stress reduction, and left ventricle (LV) function improvement have also been described [4,5,8,9,12,14,15,16,17,18,19,20].

A recent study by Li et al. [2] showed that donepezil administration in spontaneously hypertensive rats with chronic myocardial infarction also attenuated the progression of cardiovascular remodelling, improving the long-term prognosis. Additionally, the capability of the cardiomyocytes to de novo synthesise ACh and release it has been described. This cardiomyocyte cholinergic system is referred to as the “intrinsic cardiomyocyte cholinergic system” or the “non-neuronal cholinergic system”, and it represents a “positive feedback” mechanism of ACh synthesis regulation via activated muscarinic receptors in cardiomyocytes [5]. An in vitro study by Rocha-Resende et al. [21] presented evidence that this system alone can offset the effects of hyperadrenergic stimulation, while a study by Kakinuma et al. [22] showed that donepezil upregulated the choline acetyltransferase in cardiomyocytes, thus activating ACh synthesis independently of AChE inhibition or the muscarinic receptors. The underlying mechanisms of the remodelling attenuation and the distinction between the central versus peripheral effects of AChE inhibitors in this matter remain unclear. Thus, the present study aimed to investigate the effects of protracted cholinergic stimulation via the exclusively peripheral AChE inhibitor, pyridostigmine, on cardiac function and myocardial remodelling in a rat model of CHF.

## 2. Results

### 2.1. Effects of Pyridostigmine Treatment on Body Weight (BW), Heart Weight (HW), and Heart-to-Body Weight (HWBW) Ratio

There were no significant differences between groups regarding body weight, heart weight, and the heart-to-body weight ratio measured at the end of the experiment (Table 1).

### 2.2. Effects of Pyridostigmine Treatment on Acetylcholinesterase (AChE) Activity

The AChE activity was determined from erythrocyte lysate in order to confirm the direct effects of pyridostigmine on the enzyme. The administration of 22 mg/kg/day of pyridostigmine orally led to a significant decrease in AChE activity in the pyridostigmine (Pyr) and isoprenaline + pyridostigmine (Iso + Pyr) groups when compared to the Control group and isoprenaline group (Iso). The achieved level of AChE inhibition was 54.20% (Figure 1).

### 2.3. Effects of Pyridostigmine Treatment on Electrocardiogram (ECG) Recordings

A significant decrease in the heart rate was noted in the pyridostigmine-treated groups (*p* < 0.05 versus the initial HR of the same group). In the CHF groups (Iso and Iso + Pyr), a decrease in the QRS peak-to-peak voltage amplitude can be noted at the second measurement or after the CHF development. Pyridostigmine administration did not affect this parameter (Table 2).

### 2.4. Effects of Pyridostigmine Treatment on Echocardiogram (ECHO) Parameters

ECHO recordings were made at three time points: 1—initially, prior to any intervention, to determine basal values, 2—four weeks after CHF induction, to confirm CHF development, and 3—after treatment, to measure the effects. At the second time point, the isoprenaline-treated groups, Iso and Iso + Pyr, showed significant signs of cardiac damage and insufficiency. The left ventricle internal diameter end diastole (LVIDd) and end systole (LVIDs) were significantly increased in these groups (*p* < 0.001 versus Control and Pyr), with a subsequent increase in end-diastolic (EDV) and end-systolic (ESV) volumes (*p* < 0.001 versus Control and Pyr). The contractile function of the left ventricle was therefore significantly impaired, with ejection fraction (EF) values at 49.79 ± 6.18% and 43.57 ± 7.73% in the Iso and Iso + Pyr groups, respectively, compared with 79.76 ± 3.06% and 77.27 ± 7.41% in the Control and Pyr groups, respectively (*p* < 0.001) (Figure 2c,d,g,h,j). The significance of change between the first and second time point, marked as ∆2, was also determined and confirmed a significant decrease in the cardiac function of the isoprenaline-treated groups (Figure 2c,d,g,h,j, adjacent tables). In some animals, these changes were paired with noted pericardial effusion. Additionally, at this time point, the significant thinning of the posterior wall of the left ventricle was noted (Figure 2e,f and the following table). A decrease in the intraventricular septum thickness end diastole (IVSd) and end systole (IVSs) was also noted, however, without a significant change (Figure 2a,b).

At the third time point, the further progression of CHF was noted in the Iso group. This progression was significantly attenuated by pyridostigmine treatment, which prevented further dilatation (Iso LVIDd, LVIDs, EDV, and ESV *p* < 0.001 versus Iso + Pyr) and systolic dysfunction deterioration (Iso EF *p* < 0.001 versus Iso + Pyr) (Figure 2c,d,g,h,j). The further thinning of the ventricle wall was also diminished by pyridostigmine administration (Figure 2a,b,e,f). Pyridostigmine administration to healthy animals led to ventricular wall thickening (Figure 2a,b,e,f) without significance, and to significant left ventricle dilatation (Pyr LVIDd/LVIDs *p* < 0.001 and Pyr EDV/ESV *p* < 0.01 versus Control), however, without significant change in the contractile function (Pyr EF 71.42 ± 8.6%) (Figure 2j).

### 2.5. Effects of Pyridostigmine Treatment on Prooxidative and Antioxidative Markers in Myocardial Tissue Samples

Isoprenaline (ISO) administration led to a significant rise in thiobarbituric acid reactive substances (TBARS) and nitrite levels (NO_2_^−^) (*p* < 0.001 versus Control) (Figure 3a,b). Following the pyridostigmine administration, a decrease in these markers was noted, with high significance in the case of TBARS (*p* < 0.001 Iso versus Iso + Pyr). Additionally, a significant decrease in antioxidative marker catalase (CAT) and reduced glutathione (GSH) was also noted in the isoprenaline group (Figure 3c–e), which was also alleviated by pyridostigmine treatment (*p* < 0.01 Iso versus Iso + Pyr). However, regarding the measurement of superoxide dismutase (SOD), there were no significant differences found between the groups.

### 2.6. Effects of Pyridostigmine Treatment on NTproBNP, MMP-2, and MMP-9 in Serum Samples

As a marker of heart failure, the levels of N-terminal pro-brain natriuretic peptide (NTproBNP) were determined. Isoprenaline administration led to a significant increase in NTproBNP levels (*p* <0.01 compared to Control and Pyr). The administration of pyridostigmine effectively mitigated that increase (*p* < 0.01 Iso versus Iso + Pyr).

Additionally, two markers of cardiac fibrosis—matrix metalloproteinase 2 (MMP-2) and matrix metalloproteinase 9 (MMP-9)—were measured. In the isoprenaline-treated group (Iso), the levels of both enzymes were significantly elevated (*p* < 0.05 and *p* <0.001, respectively, versus Control). Treatment with pyridostigmine prevented this increase (*p* < 0.05 Iso versus Iso + Pyr) (Table 3).

### 2.7. Effects of Pyridostigmine Treatment on Histological Structure of Cardiac Muscle, H&E Stain, Masson’s Trichrome Stain, and Tissue Damage Score

Extensive myocardial damage, with extensive fibrosis and cardiomyocyte damage, was noted in the Iso group, while in the Iso + Pyr group, largely preserved tissue architecture was noted. A significant reduction in the tissue damage score was noted, thus demonstrating an alleviation of the isoprenaline-induced damage by the pyridostigmine administration. Additionally, in the Pyr group, a mild hypertrophy of cardiomyocytes was noted (Figure 4).

The extent of myocardial fibrosis in the Iso group was also confirmed in Masson’s trichrome stain, which distinctly marks collagen deposition within the myocardial tissue. Pyridostigmine administration led to cardiac tissue architecture preservation and a decrease in cardiac collagen deposition (Figure 5).

## 3. Discussion

The present study used an isoprenaline model of chronic heart failure (CHF) to investigate the cardioprotective potential of pyridostigmine, a peripheral acetylcholinesterase (AChE) inhibitor. Chronic AChE inhibition with pyridostigmine prevented the progression of CHF, decreasing chamber wall thinning (↑ PWDd, ↑ PWDs) and dilatation progression (↓ LVIDd, ↓ LVIDs), thus improving cardiac contractile function (↑ EF), which was coupled with a decrease in the serum CHF marker NTproBNP. Additionally, pyridostigmine improved antioxidative status (↓ TBARS, ↓ NO_2_^−^; ↑ CAT, ↑ GSH) and significantly reduced cardiac fibrosis development, confirmed by pathohistological findings and biochemical marker reduction (↓ MMP2, ↓ MMP9).

Pyridostigmine acts as a reversible peripheral AChE inhibitor, increasing the concentrations of acetylcholine in the synaptic clefts throughout the body. By increasing the ACh concentrations, an enhancement of vagal efferent effects is achieved, which is considered beneficial in the case of CHF, where an autonomic imbalance with an increased sympathetic and decreased parasympathetic tone is described. In addition, heart rate is clinically recognised as an independent prognostic factor in CHF, as heart rate reduction in patients with heart failure is associated with survival improvement [23]. By increasing ACh levels, pyridostigmine supports the parasympathetic negative chronotropic effects, decreasing the heart rate. This effect was shown in the present study, where the basal heart rate recorded via ECG was significantly reduced in the pyridostigmine-treated groups, in which AChE activity was significantly reduced. Although not investigated in the present study, previous studies have shown additional beneficial effects of pyridostigmine on heart rate variability (HRV) [1]. In spontaneously hypertensive rats, pyridostigmine administration increased the total HRV, especially its very low frequency (VLF) component, and attenuated the HR increase during stress, indicating increased vagal efferent effects [1]. Similar effects were described with a combined peripheral and central AChE inhibitor, donepezil; however, the effects were less pronounced, thus indicating a possible greater importance of the peripheral AChE inhibition [1]. Another study, by Li et al. [4], demonstrated that donepezil treatment led to a significant decrease in the average heart rate but without causing a decrease in blood pressure in a rat model of CHF. Knowing that β-blockers, the gold standard for CHF treatment, express substantial negative inotropic effects, AChE inhibitors could be more beneficial for patients with decompensated CHF [4].

In the present study, pyridostigmine treatment significantly attenuated the progression of isoprenaline-induced CHF. Pyridostigmine treatment preserved the systolic function, which was in concordance with previous studies by Lataro et al. [8] and Feriani et al. [20], who investigated the effects of pyridostigmine administration combined with aerobic exercise as an early intervention after myocardial infarction in rats. Interestingly, pyridostigmine administration to healthy animals (Pyr group) led to ventricular wall thickening and an increase in the left ventricle diameter end diastole and end systole, however, without a significant change in the contractile function. As previously mentioned, the basal heart rate in this group was reduced, also, mild hypertrophy of cardiomyocytes in this group was noted in the histological preparations. Similarly, previous studies have reported that pyridostigmine administration in healthy humans leads to bradycardia, without functional impairment [8,24,25,26]. These findings can be explained by the compensatory hypertrophy of cardiomyocytes as a response to an increased vagal tone.

Today, multiple pathogenic mechanisms are known to be involved in CHF development, including hemodynamic overload, ventricular structural changes, and dysfunction secondary to ischemia [27]. Oxidative stress and inflammation have been recognised as crucial elements in the pathophysiology of heart failure, in which the upregulation of inflammatory mediators such as cytokines, chemokines, and adhesion molecules leads to disease progression [28]. During myocardial ischaemia, an increase in reactive oxygen and nitrogen species (ROS/RNS) production and inflammatory cytokine generation not only leads to cardiomyocyte apoptosis but also activates an inflammatory response, recruiting inflammatory cells to the ischaemic area. This further promotes the generation of free radicals, causing a self-perpetuating cycle. This cross-talk between oxidative stress and inflammation enhances adverse cardiac remodelling and progressive heart failure [5,27,29,30]. Results from the present study show that pyridostigmine has the potential to interrupt this pathophysiological mechanism and improve cardiac oxidative status and attenuate cardiac remodelling, reducing the fibrosis of the cardiac muscle. This was shown both in the pathohistological investigations and by the reduction in the MMP-2 and MMP-9 levels. MMP expression has been shown to increase both in the initial and the final phase of heart failure [31]. Additionally, in patients with dilated cardiomyopathy, an increase in MMP expression is associated with decreased cardiac function [31,32]. Previous studies have shown that pyridostigmine administration, acutely after a myocardial infarction, enhances recruitment of anti-inflammatory cells, increasing the M2 macrophage and FOXP3+ cell count in the infarcted area [9,16]. This is one of the possible mechanisms of beneficial pyridostigmine effects.

The interplay between the autonomic nervous system and immune system has gained attention as a potent novel therapeutic approach in the treatment of several different diseases, including sepsis, arthritis, and other inflammatory conditions [33,34,35]. Named as the cholinergic anti-inflammatory pathway (CAP), this sophisticated neuroimmune axis regulates the crosstalk between the nervous system and the immune response via the vagal nerve [36]. The process is believed to be mediated by nicotinic cholinergic receptors in the immune cells [9]. The effects of pyridostigmine on this axis in the case of CHF have not yet been fully investigated. Donepezil administration in CHF rats has been shown to reduce pro-inflammatory markers, like TNF-α, indicating the connection between the beneficial effects of AChE inhibitors and CAP in chronic heart failure [4]. The intracellular mechanisms of CAP are, however, still under investigation. Previous reports indicate that cholinergic stimulation with acetylcholine and nicotinic receptor agonists inhibits LPS-induced activation of NF-κB in macrophages via the α7 nicotinic receptor (α7nAChR) [37]. Another study reported the cardioprotective effects of α7nAChR agonist PNU (PNU-282987), which upregulated mitochondrial fusion and attenuated autophagy, and muscarinic (mAChR) receptor agonist bethanechol, which attenuated mitochondrial fission and mitophagy [38]. Vagal nerve stimulation and donepezil administration have also shown the ability to attenuate inflammation by inhibiting NF-κB in macrophages in a rat model of CHF [9,17,39]. However, data about the mechanisms of the pyridostigmine anti-inflammatory effects in CHF, although possibly similar to the ones previously described, are scarce. Previous reports show that pyridostigmine interferes positively with several intracellular pathways; i.e., Lu et al. [18] demonstrated that pyridostigmine administration lowers the activation of the TGFβ1/TAK1 pathway, which is directly involved in cardiac hypertrophy and fibrosis. A more recent study by Xue et al. [40] reported the pyridostigmine alleviation of obesity-induced hepatic injury in mice via the α7nAChR and M3 muscarinic receptor (M3AchR) by mitigating mitochondrial damage and oxidative stress. The proposed mechanism could be translated to the CHF model.

## 4. Materials and Methods

### 4.1. Experimental Animals and Experimental Protocol

Male Wistar albino rats (n = 24) weighing 300 ± 50 g were used in this experiment. Animals were kept under controlled laboratory conditions, at 21 ± 2 °C room temperature, 55 ± 5% humidity, and a 12 h light–dark cycle. They were given access to food and water ad libitum. They received a standard pellet diet purchased from the Veterinary Institute of Subotica (Subotica, Serbia). After obtaining basal ECG and echocardiogram (ECHO) measurements, animals were randomised into four groups. Animals in the isoprenaline group (Iso; n = 8) and isoprenaline + pyridostigmine group (Iso + Pyr; n = 8) were treated with 5 mg/kg/day of isoprenaline (ISO) and dissolved in normal saline, s.c., for 7 days, while animals in the Control group (Control; n = 6) and the pyridostigmine group (Pyr, n = 6) received an equivalent amount of normal saline, s.c. Animals were then followed for the next 4 weeks, after which a second round of ECG and ECHO measurements were obtained to confirm the development of chronic heart failure [41,42]. For the next 14 days, rats in the Pyr and Iso + Pyr groups received 22 mg/kg/day of pyridostigmine dissolved in drinking water (0.14 mg/mL), while rats in the Control and Iso groups were untreated. At the end of the experiment, after obtaining the final ECG and ECHO measurements, animals were anaesthetised using a combination of 90 mg/kg of ketamine and 10 mg/kg of xylazine and then sacrificed by exsanguination, and the tissue and blood samples were collected (Figure 6).

### 4.2. ECG Recording and Echocardiogram (ECHO)

ECG recordings and ECHO were conducted on three separate occasions—at the beginning of the experiment to establish the basal values, 4 weeks post-isoprenaline administration to confirm CHF development, and at the end of the experiment. Before each of the recordings, the rats were anaesthetised using a combination of ketamine (30 mg/kg) and xylazine (5 mg/kg) [43]. The ECGs were captured with a sensitivity setting of 2 cm per 1 mV and a 25 mm/sec speed of paper. Lead II was used to analyse the heart rate (beats per minute, bpm), PQ/PR interval (seconds, s), QT/RT interval (seconds, s), and QRS peak-to-peak voltage amplitude (millivolts, mV) in each tracing [44]. After the ECG, transthoracic two-dimensional (2D) echocardiography was performed using a Logiq 400 CL ultrasound device (GE Medical Systems, Chicago, IL, USA) with an 11 MHz phased array transducer to assess cardiac structure and function. M-mode echocardiographic images were captured in parasternal long-axis and short-axis views at the papillary muscle tips. Measurements included systolic and diastolic septal (IVSs and IVSd), posterior wall thickness (PDWs and PDWd), and left ventricular internal diameters (LVIDs and LVIDd). The left ventricular ejection fraction (EF), end-systolic volume (ESV), end-diastolic volume (EDV), and stroke volume (SV) were calculated [45]. All measurements were conducted by the same observer following the American Society of Echocardiography guidelines [46,47].

### 4.3. Acetylcholinesterase (AChE) Activity Measurement

The AChE activity was measured by the Ellman colorimetric method using a Shimadzu UV-1800 spectrophotometer (Kyoto, Japan) and UV Probe 2.17 software (Kyoto, Japan). Activity was determined from blood samples (erythrocyte lysate) and expressed as U/L.

### 4.4. NTproBNP, MMP-2, and MMP-9

N-terminal pro-brain natriuretic peptide (NT-proBNP), matrix metalloproteinase 2 (MMP-2), and matrix metalloproteinase 9 (MMP-9) were measured in serum by an enzyme-linked immunosorbent assay (ELISA) using the FineTest^®^ (Wuhan Fine Biotech Co., Ltd., Wuhan, China) Rat NT-proBNP ELISA Kit, Rat MMP-2 ELISA Kit, and Rat MMP-9 ELISA Kit, respectively. Assays were performed according to the manufacturer’s recommendations.

### 4.5. Hearth Tissue Homogenisation

After excision, rat hearts were rinsed in ice-cold normal saline and frozen at −20 °C. The tissue homogenate was prepared in ice-cold phosphate buffer (pH 7.4) using an HG-15D homogeniser (Witeg labortechnik GmbH, Wertheim, Germany) and centrifuged at +4 °C and 1200× *g*. The supernatant was used to determine the levels of prooxidative and antioxidative markers.

### 4.6. Prooxidative and Antioxidative Markers

For the determination of nitrite levels (NO_2_^−^), the Griess method was employed, using the Griess reagent. After colour stabilisation at room temperature for 5–10 min, the concentration of released nitrites was measured spectrophotometrically at a wavelength of λ = 550 nm [48,49]. The lipid peroxidation index, or thiobarbituric acid reactive substances (TBARS), was determined indirectly, using 1% TBA and 0.05 M sodium hydroxide (NaOH) and measured spectrophotometrically at 530 nm [50]. Antioxidative markers—CAT, SOD, and GSH—were measured spectrophotometrically using Beutler methods [51,52].

### 4.7. Histopathological Analysis

After dissection, isolated rat hearts were fixed in 4% formaldehyde for 48 h to ensure the optimal preservation of tissue morphology. Afterwards, the samples were moulded into blocks with paraffin wax and cut into 4 μm slices using a standard-issue microtome. The slices were then stained with haematoxylin (Fisher Scientific, Geel, Belgium) and eosin (Sigma Aldrich, St. Louise, MO, USA) dye (H&E). An analysis was performed using a Leica DM 6000 binocular microscope equipped with a Leica DFC310FX camera (Leica Microsystems, Mannheim, Germany). Cardiac muscle damage was assessed using a semiquantitative scoring system based on established methodology [44,51,53]. An analysis was conducted on ten random visual fields at 20× magnification by a histologist blinded to the treatment group labels to ensure unbiased evaluation. Each slice was scored from 1 to 4, and an average group score was calculated. A score of 1 indicates no pathological changes in the myocardium; 2—mild focal damage of cardiomyocytes, including myofibrillar hypertrophy with mild inflammatory infiltrates; 3—moderate damage, >50% of cardiomyocytes characterised by myofibrillar volume fraction changes and vacuolar degeneration; 4—severe changes, including extensive tissue fibrosis. The assessment of myocardial fibrosis is performed by Masson’s trichrome stain (Cellavision Kit Trichrome de Masson), which distinctly marks the deposition of collagen within the myocardial tissue.

### 4.8. Statistical Analysis

Statistical analysis was conducted using IBM-SPSS Statistics version 17.0 software (SPSS, Inc., Chicago, IL, USA). The Kruskal–Wallis and Mann–Whitney U tests were used to compare the nonparametric characteristics between the groups, followed by Tukey and Bonferroni tests for post hoc analysis. The results are expressed as mean ± standard error, with a *p*-value of less than 0.05 deemed statistically significant.

## 5. Conclusions

In the present study, chronic peripheral AChE inhibition via pyridostigmine showed compelling antioxidative, anti-inflammatory, and antifibrotic effects, with the subsequent preservation of cardiac functional parameters in a rat model of CHF. However, further investigations are needed to fully understand the exact cellular mechanisms involved in the CHF attenuation via pyridostigmine.

## Figures and Tables

**Figure 1 ijms-26-06892-f001:**
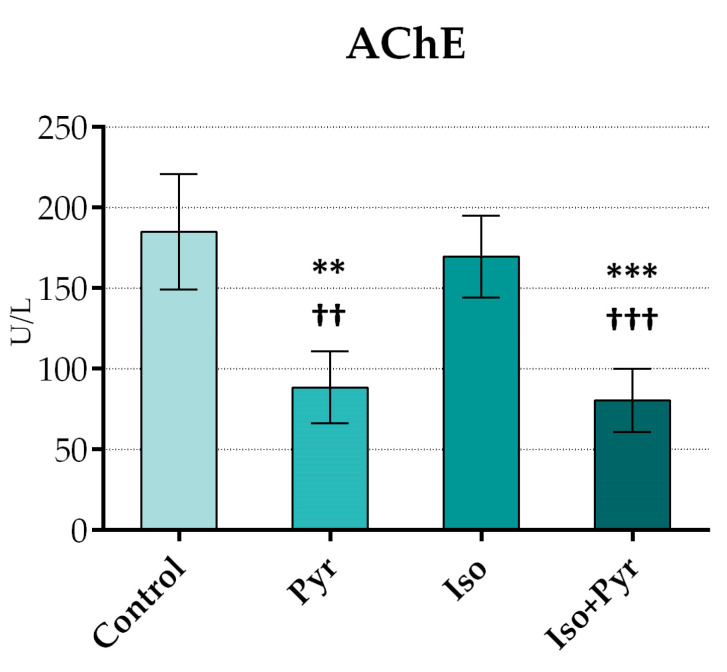
Effects of pyridostigmine treatment on acetylcholinesterase (AChE) activity. Control—0.9% NaCl s.c. for 7 days + 28 days observation + tap water p.o. 14 days; Pyr—0.9% NaCl s.c. for 7 days + 28 days observation + 22 mg/kg/day of pyridostigmine p.o. 14 days; Iso—5 mg/kg/day of isoprenaline s.c. for 7 days + 28 days observation + tap water p.o. 14 days; Iso + Pyr—5 mg/kg/day of isoprenaline s.c. for 7 days + 28 days observation + 22 mg/kg/day of pyridostigmine p.o. 14 days; ** *p* < 0.01 versus Control; *** *p* < 0.001 versus Control; ^††^ *p* < 0.01 versus Iso; ^†††^
*p* < 0.001 versus Iso.

**Figure 2 ijms-26-06892-f002:**
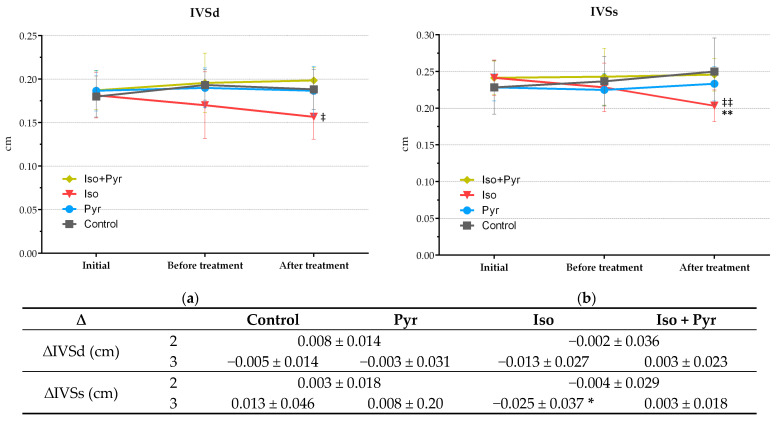
Effects of pyridostigmine treatment on echocardiogram (ECHO) parameters. (**a**) IVSd—interventricular septum thickness end diastole; (**b**) IVSs—interventricular septum thickness end systole; (**c**) LVIDd—left ventricle internal diameter end diastole; (**d**) LVIDs—left ventricle internal diameter end systole; (**e**) PWDd—posterior wall diameter end diastole; (**f**) PWDs—posterior wall diameter end systole; (**g**) EDV—end-diastolic volume; (**h**) ESV—end-systolic volume; (**i**) SV—stroke volume; (**j**) EF—ejection fraction; ∆IVSd/IVSs—interventricular septum thickness end diastole and end systole change; ∆LVIDd/LVIDs—left ventricle internal diameter end diastole and end systole change; ∆PWDd/PWDs—posterior wall diameter end diastole and end systole change; ∆EDV—end-diastolic volume change; ∆ESV—end-systolic volume change; ∆SV—stroke volume change; ∆EF—ejection fraction change; 2—between first and second recording, after CHF induction; 3—between second and third recording, after treatment; Control—0.9% NaCl s.c. for 7 days + 28 days observation + tap water p.o. 14 days; Pyr—0.9% NaCl s.c. for 7 days + 28 days observation + 22 mg/kg/day of pyridostigmine p.o. 14 days; Iso—5 mg/kg/day of isoprenaline s.c. for 7 days + 28 days observation + tap water p.o. 14 days; Iso + Pyr—5 mg/kg/day of isoprenaline s.c. for 7 days + 28 days observation + 22 mg/kg/day of pyridostigmine p.o. 14 days; * *p* < 0.05 versus Control at the same time point, ** *p* < 0.01 versus Control at the same time point, *** *p* < 0.001 versus Control at the same time point, ^†^ *p* < 0.05 versus Pyr at the same time point, ^††^ *p* < 0.01 versus Pyr at the same time point, ^†††^ *p* < 0.001 versus Pyr at the same time point, ^‡^ *p* < 0.05 versus Iso + Pyr at the same time point, ^‡‡^ *p* < 0.01 versus Iso + Pyr at the same time point, and ^‡‡‡^ *p* < 0.001 versus Iso + Pyr at the same time point. All values are expressed as mean ± SD.

**Figure 3 ijms-26-06892-f003:**
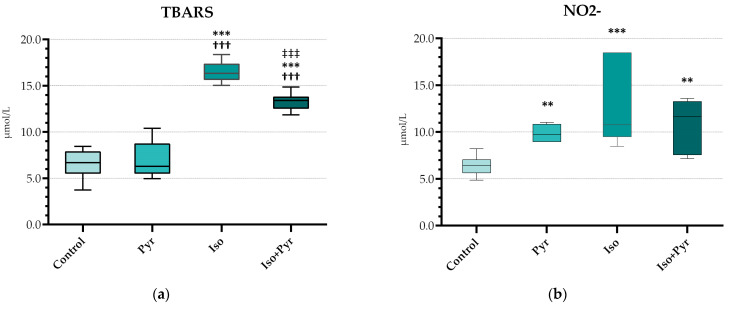
Effects of pyridostigmine treatment on prooxidative and antioxidative markers in myocardial tissue samples. (**a**) Thiobarbituric acid reactive substances (TBARS); (**b**) nitrite levels (NO_2_^−^); (**c**) superoxide dismutase (SOD); (**d**) reduced glutathione (GSH); (**e**) catalase (CAT); Control—0.9% NaCl s.c. for 7 days + 28 days observation + tap water p.o. 14 days; Pyr—0.9% NaCl s.c. for 7 days + 28 days observation + 22 mg/kg/day of pyridostigmine p.o. 14 days; Iso—5 mg/kg/day of isoprenaline s.c. for 7 days + 28 days observation + tap water p.o. 14 days; Iso + Pyr—5 mg/kg/day of isoprenaline s.c. for 7 days + 28 days observation + 22 mg/kg/day of pyridostigmine p.o. 14 days; * *p* < 0.05 versus Control; ** *p* < 0.01 versus Control; *** *p* < 0.001 versus Control; ^††^ *p* < 0.01 versus Pyr, ^†††^ *p* < 0.001 versus Pyr, ^‡‡^ *p* < 0.01 versus Iso, and ^‡‡‡^ *p* < 0.001 versus Iso.

**Figure 4 ijms-26-06892-f004:**
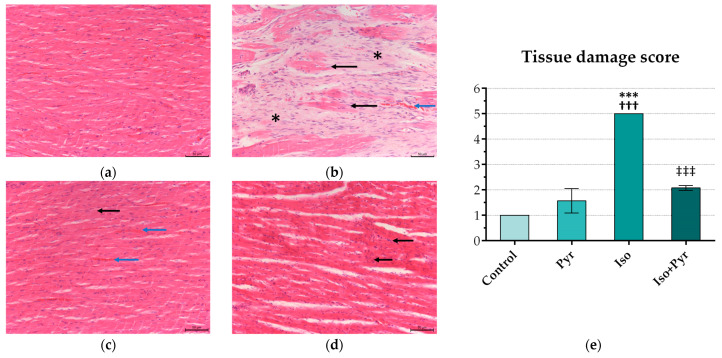
Effects of pyridostigmine treatment on histological structure of cardiac muscle, septal region of the heart, H&E staining magnification ×200, scale bar 50 µm. (**a**) Control—normal histological structure of cardiac muscle; (**b**) Iso—extensive myocardial fibrosis (asterisk), cardiomyocyte damage (black arrow), and erythrocyte extravasation in the endomysium (blue arrow); (**c**) Pyr—mild hypertrophy of cardiomyocytes is noted (black arrow), with erythrocyte extravasation in the endomysium (blue arrow); (**d**) Iso + Pyr—largely preserved tissue architecture, with an increased number of inflammatory cells and fibroblasts observed in the endomysium (black arrow). (**e**) Effects of pyridostigmine treatment on the tissue damage score. Control—0.9% NaCl s.c. for 7 days + 28 days observation + tap water p.o. 14 days; Pyr—0.9% NaCl s.c. for 7 days + 28 days observation + 22 mg/kg/day of pyridostigmine p.o. 14 days; Iso—5 mg/kg/day of isoprenaline s.c. for 7 days + 28 days observation + tap water p.o. 14 days; Iso + Pyr—5 mg/kg/day of isoprenaline s.c. for 7 days + 28 days observation + 22 mg/kg/day of pyridostigmine p.o. 14 days. *** *p* < 0.001 versus Control; ^†††^ *p* < 0.001 versus Pyr; ^‡‡‡^ *p* < 0.001 versus Iso.

**Figure 5 ijms-26-06892-f005:**
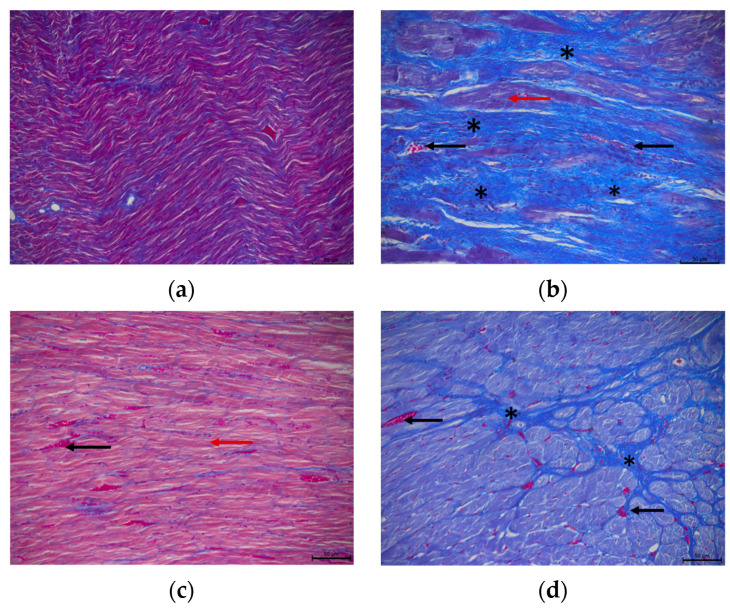
Effects of pyridostigmine treatment on histological structure of cardiac muscle, septal region of the heart, Masson’s trichrome stain, magnification ×200, scale bar 50 µm. (**a**) Control—representative image showing the normal histological structure of cardiac muscle. Cardiomyocytes are stained red; (**b**) Iso—extensive myocardial fibrosis is observed, with fibrotic areas stained blue (asterisk). Cardiomyocyte damage is indicated (red arrow), along with erythrocyte extravasation in the endomysium (black arrow); (**c**) Pyr—cardiomyocytes are stained red, with mild hypertrophy noted (red arrow). Erythrocyte extravasation is present in the endomysium (black arrow); (**d**) Iso + Pyr—largely preserved tissue architecture with erythrocyte extravasation in the endomysium (black arrow). Fibrosis is present in a small area, with collagen fibres stained blue (asterisk). Dilated capillaries are visible in the endomysium. Control—0.9% NaCl s.c. for 7 days + 28 days observation + tap water p.o. 14 days; Pyr—0.9% NaCl s.c. for 7 days + 28 days observation + 22 mg/kg/day pyridostigmine p.o. 14 days; Iso—5 mg/kg/day isoprenaline s.c. for 7 days + 28 days observation + tap water p.o. 14 days; Iso + Pyr—5 mg/kg/day isoprenaline s.c. for 7 days + 28 days observation + 22 mg/kg/day pyridostigmine p.o. 14 days.

**Figure 6 ijms-26-06892-f006:**
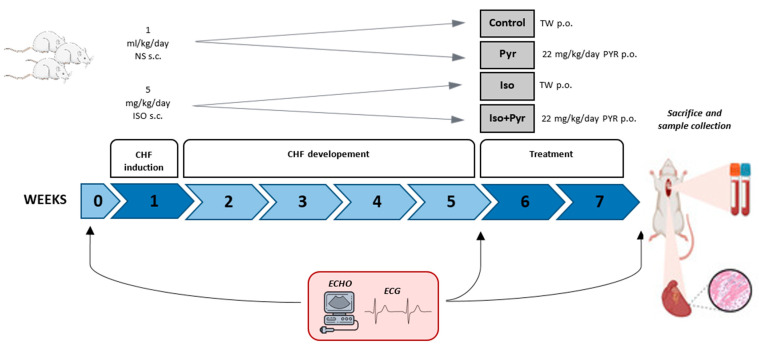
Study design. Control group (Control; n = 6); pyridostigmine group (Pyr; n = 6); isoprenaline group (Iso; n = 8); pyridostigmine group (Iso + Pyr; n = 8); NS—normal saline; ISO—isoprenaline; TW—tap water; PYR—pyridostigmine; CHF—chronic heart failure.

**Table 1 ijms-26-06892-t001:** Effects of pyridostigmine treatment on body weight (BW), heart weight (HW), and heart-to-body weight (HW/BW) ratio.

	HW (mg)	BW (g)	HW/BW Ratio (mg/g)
Control	1340.00 ± 330.00	409.67 ± 38.05	3.26 ± 0.75
Pyr	1400.00 ± 310.00	390.00 ± 20.14	3.58 ± 0.81
Iso	1270.00 ± 260.00	409.71 ± 65.86	3.10 ± 0.49
Iso + Pyr	1340.00 ± 60.00	396.50 ± 25.73	3.40 ± 0.22

HW—heart weight; BW—body weight; HW/BW ratio—heart-to-body weight ratio; Control—0.9% NaCl s.c. for 7 days + 28 days observation + tap water p.o. 14 days; Pyr—0.9% NaCl s.c. for 7 days + 28 days observation + 22 mg/kg/day of pyridostigmine p.o. 14 days; Iso—5 mg/kg/day of isoprenaline s.c. for 7 days + 28 days observation + tap water p.o. 14 days; Iso + Pyr—5 mg/kg/day of isoprenaline s.c. for 7 days + 28 days observation + 22 mg/kg/day of pyridostigmine p.o. 14 days. All values are expressed as mean ± SD.

**Table 2 ijms-26-06892-t002:** Effects of pyridostigmine treatment on electrocardiogram (ECG) recordings.

		Control	Pyr	Iso	Iso + Pyr
HR (bpm)	1	244.09 ± 28.88	258.27 ± 23.73	261.85 ± 16.99	268.18 ± 18.37
	2	246.44 ± 20.99	259.50 ± 27.32	244.13 ± 29.39	257.12 ± 17.24
	3	246.29 ± 20.99	224.67 ± 16.96 *****^†^	249.12 ± 23.24	231.34 ± 12.12 ******^†^
PQ/PR (ms)	1	49.17 ± 8.01	46.67 ± 8.16	45.00 ± 5.35	44.38 ± 4.96
	2	50.00 ± 7.07	47.50 ± 5.00	44.29 ± 5.35	49.17 ± 6.65
	3	50.00 ± 7.07	46.67 ± 5.77	48.57 ± 6.90	43.33 ± 5.16
QT/RT interval (ms)	1	86.67 ± 5.16	94.17 ± 10.21	101.25 ± 9.91	93.75 ± 5.18
	2	96.67 ± 10.33	103.33 ± 8.16	107.14 ± 9.51	104.29 ± 9.76
	3	96.67 ± 10.33	95.00 ± 10.49	102.86 ± 7.56	95.71 ± 7.87
QRS peak-to-peak voltage amplitude (mV)	1	0.26 ± 0.04	0.25 ± 0.03	0.26 ± 0.05	0.26 ± 0.02
	2	0.27 ± 0.07	0.28 ± 0.06	0.23 ± 0.06	0.20 ± 0.03 ^‡‡^
	3	0.27 ± 0.07	0.25 ± 0.07	0.24 ± 0.06	0.20 ± 0.03 ^‡‡‡^

HR—heart rate; bpm—beats per minute; PQ/PR—PQ/PR interval; QT/RT—QT/RT interval; QRS—QRS peak-to-peak voltage amplitude; 1—measured on the first recording, prior to any intervention; 2—measured on the second recording, after CHF induction; 3—measured on the third recording, after treatment; Control—0.9% NaCl s.c. for 7 days + 28 days observation + tap water p.o. 14 days; Pyr—0.9% NaCl s.c. for 7 days + 28 days observation + 22 mg/kg/day of pyridostigmine p.o. 14 days; Iso—5 mg/kg/day of isoprenaline s.c. for 7 days + 28 days observation + tap water p.o. 14 days; Iso + Pyr—5 mg/kg/day of isoprenaline s.c. for 7 days + 28 days observation + 22 mg/kg/day of pyridostigmine p.o. 14 days; * *p* < 0.05 versus HR1 of the same group, ** *p* < 0.01 versus HR1 of the same group, and ^†^ *p* < 0.05 versus HR2 of the same group; ^‡‡^ *p* < 0.01 versus QRS1 of the same group, and ^‡‡‡^ *p* < 0.001 versus QRS1 of the same group. All values are expressed as mean ± SD.

**Table 3 ijms-26-06892-t003:** Effects of pyridostigmine treatment on NTproBNP, MMP-2, and MMP-9 in serum samples.

	NTproBNP	MMP-2	MMP-9
Control	346.00 ± 44.66	134.70 ± 18.57	47951.42 ± 9486.37
Pyr	366.19 ± 26.26	212.96 ± 82.17	49905.53 ± 10813.50
Iso	517.72 ± 52.24 ****** ^††^	307.04 ± 54.50 ******	59619.24 ± 4502.97 *****
Iso + Pyr	383.15 ± 50.36 ^‡‡^	203.09 ± 70.22 ^‡^	52025.53 ± 5861.58 ^‡^

NTproBNP—N-terminal pro-brain natriuretic peptide; MMP-2—matrix metalloproteinase 2; MMP-9—matrix metalloproteinase 9; Control—0.9% NaCl s.c. for 7 days + 28 days observation + tap water p.o. 14 days; Pyr—0.9% NaCl s.c. for 7 days + 28 days observation + 22 mg/kg/day of pyridostigmine p.o. 14 days; Iso—5 mg/kg/day of isoprenaline s.c. for 7 days + 28 days observation + tap water p.o. 14 days; Iso + Pyr—5 mg/kg/day of isoprenaline s.c. for 7 days + 28 days observation + 22 mg/kg/day of pyridostigmine p.o. 14 days; * *p* < 0.05 versus Control; ** *p* < 0.01 versus Control; ^††^ *p* < 0.01 versus Pyr, ^‡^ *p* < 0.05 versus Iso, and ^‡‡^ *p* < 0.01 versus Iso.

## Data Availability

The data presented in this study are available on request from the corresponding author.

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
