# Peer review of "Pyridostigmine Treatment Significantly Alleviates Isoprenaline-Induced Chronic Heart Failure in Rats"

_ijms, 2025, doi:10.3390/ijms26146892_

Round 1
Reviewer 1 Report
Comments and Suggestions for Authors
Marinković and colleagues present an interesting paper showing that administration of pyridostigmine significantly improves is isoproterenol-induced chronic heart failure in rats. In this carefully performed study, administration of pyridostigmine resulted in attenuation of the left ventricular dilatation and wall thinning, improvement of left ventricle systolic function, alleviation of oxidative stress, and attenuation of cardiac fibrosis.
This is a well-executed and important study that advances our understanding of the role of the parasympathetic nervous system in heart failure. I have only a few comments:
- The authors indicate that pyridostigmine is a peripheral AChE inhibitor. However, in their conclusions, they indicate that further research is needed to differentiate between central and peripheral effects of a AChE inhibition. Since pyridostigmine acts peripherally, would the present data not indicate that the beneficial effects observed are due to peripheral AChE inhibition?
- Please indicate in the text and figure legends where the measurements of TBARS, NO2, SOD, GSH, CAT, BNP, MMP2, and MMP9 were performed. Presumably, some of these assays were done in plasma or serum and some in myocardial tissue. Please specify this in the text and in each figure legend.
- Besides the micrographs shown in Fig. 5, it would be interesting to quantify myocardial collagen content, as assessed by Masson‘s trichrome stain.
Author Response
Comment 1: “The authors indicate that pyridostigmine is a peripheral AChE inhibitor. However, in their conclusions, they indicate that further research is needed to differentiate between central and peripheral effects of AChE inhibition. Since pyridostigmine acts peripherally, would the present data not indicate that the beneficial effects observed are due to peripheral AChE inhibition?”
Response 1: Thank you for this interesting question. We agree with the reviewer that since pyridostigmine acts peripherally, the beneficial effects of its administration observed in this study are due to peripheral AChE inhibition. In the conclusions, we have written that “…further investigations are needed to fully differentiate between central and peripheral AChE inhibition effects and the exact cellular mechanisms involved in the CHF attenuation via pyridostigmine.” This sentence regards the differentiation between specific cellular mechanisms of central AChE inhibitors versus the mechanisms of the beneficial effects of peripheral AChE inhibitors. The question is whether they are the same or they differ in some matter. As it was mentioned in the introduction (lines 93 – 102), previous studies reported that some beneficial effects of AChE inhibitors were independent of AChE inhibition or the muscarinic receptors. However, we agree with the reviewer that the sentence used in the conclusions is not clear enough, so we have made changes to the text accordingly. The changes can be found on page no. 1, line 48 and page no. 16, line 456.
Comment 2: “Please indicate in the text and figure legends where the measurements of TBARS, NO2, SOD, GSH, CAT, BNP, MMP2, and MMP9 were performed. Presumably, some of these assays were done in plasma or serum and some in myocardial tissue. Please specify this in the text and in each figure legend.”
Response 2: We have made changes to the article text and figure legends, specifying the sample source for all of the above-mentioned measurements. The changes can be found on page no. 8, line 192-193, page no. 9, lines 204, 211 and 222, and page no. 14, line 408.
Comment 3: “Besides the micrographs shown in Fig. 5, it would be interesting to quantify myocardial collagen content, as assessed by Masson‘s trichrome stain.”
Response 3: We agree with the reviewer that quantification of the myocardial collagen content would provide significant data and strengthen the proof of the study. We will be planning to do this quantification alongside other additional histological investigations, such as immunohistochemistry, in order to further investigate the effects of pyridostigmine.
Reviewer 2 Report
Comments and Suggestions for Authors
The work presented is very interesting the data are well presented. The role of cholinergic singling in the heart and the work is clinically important.
The question arises is what is the relationship between the cholinergic pathway and the adrenergic pathway in this model could the authors please provide a brief comment on the interaction between these important cycling pathways and the heart it would also be interesting to know whether these effects are reversible in terms of fibrosis and cardiac function. Please comment accordingly.
Author Response
Comment 1: “The question arises is what is the relationship between the cholinergic pathway and the adrenergic pathway in this model could the authors please provide a brief comment on the interaction between these important cycling pathways and the heart it would also be interesting to know whether these effects are reversible in terms of fibrosis and cardiac function. Please comment accordingly.”
Response 1: Regarding the interaction of the adrenergic pathway of the isoprenaline model and the cholinergic pathway of the pyridostigmine treatment, we consider that there are no direct interactions between the two. Isoprenaline is used only as a mechanism of myocardial damage, in a similar fashion as one would use the LAD ligation model. It is administered at the beginning of the protocol, and we consider all further sympathetic overdrive changes that are described to be a compensatory mechanism in the chronic heart failure itself, and not a direct effect of isoprenaline.
As for the reversibility of the fibrosis and cardiac function, since the model used is a model of chronic heart failure, we propose that already existing fibrosis is not reversed, but rather that the pyridostigmine treatment prevents further damage that would develop as a consequence of the chronic heart failure. Thus, less fibrotic tissue is noted in the Isoprenaline + Pyridostigmine group when compared to the isoprenaline group.
Reviewer 3 Report
Comments and Suggestions for Authors
By employing an isoproterenol- induced rat model of heart failure, the authors have shown that the progression of contractile dysfunction is attenuated by treatment with Pyridostigmine. The beneficial effects of this AchE inhibitors on cardiac remodelling were shown to be associated with the reduction of oxidative stress, reduced cardiac fibrosis and depressed activities of both MMP2 AND MMP9. It is an excellent piece of work of some fundamental importance. I have no comments for any improvement in this paper and thus pleased to recommend acceptance for publication.
Author Response
Comment 1: “I have no comments for any improvement in this paper and thus pleased to recommend acceptance for publication.”
Response 1: Thank you for your comments and recognition of our work.